# Patterns of Endemism in Lichens: Another Paradigm-Shifting Example in the Lichen Genus *Xanthoparmelia* from Macaronesia

**DOI:** 10.3390/jof10030166

**Published:** 2024-02-21

**Authors:** Israel Pérez-Vargas, Javier Tuero-Septién, Nereida M. Rancel-Rodríguez, José Antonio Pérez, Miguel Blázquez

**Affiliations:** 1Department of Botany, Ecology and Plant Physiology, Faculty of Pharmacy, University of La Laguna, Apdo Postal 456, 38200 La Laguna, Canary Islands, Spainnrrodri@ull.edu.es (N.M.R.-R.); 2Instituto Universitario de Enfermedades Tropicales y Salud Pública de Canarias, Área de Genética, Universidad de La Laguna, Apdo Postal 456, 38200 La Laguna, Canary Islands, Spain; joanpere@ull.edu.es; 3Department of Mycology, Real Jardín Botánico (CSIC), 28014 Madrid, Madrid, Spain

**Keywords:** lichens, Canary Islands, biogeography, disjunctions, new species

## Abstract

It has long been assumed that lichen-forming fungi have very large distribution ranges, and that endemic species are rare in this group of organisms. This is likely a consequence of the “everything small is everywhere” paradigm that has been traditionally applied to cryptogams. However, the description of numerous endemic species over the last decades, many of them in oceanic islands, is challenging this view. In this study, we provide another example, *Xanthoparmelia ramosae*, a species that is described here as new to science on the basis of morphological, chemical, and macroclimatic data, and three molecular markers (*ITS* rDNA, *nuLSU* rDNA, and *mtSSU*). The new species is endemic to the island of Gran Canaria but clusters into a clade composed exclusively of specimens collected in Eastern Africa, a disjunction that is here reported for the first time in lichen-forming fungi. Through the use of dating analysis, we have found that *Xanthoparmelia ramosae* diverged from its closely related African taxa in the Pliocene. This result, together with the reproductive strategy of the species, points to the Relict theory as a likely mechanism behind the disjunction, although the large gap in lichenological knowledge in Africa makes this possibility hard to explore any further.

## 1. Introduction

Lichens are, by definition, symbiotic organisms typically composed of a fungus (the mycobiont) and one or more photosynthetic organisms (the photobionts), which may be green algae and/or cyanobacteria [1,2,3]. Historically, it was widely believed that these organisms had broad distribution ranges, and the presence of endemic species was considered rare [4]. This is probably a result of the long-standing application of the paradigm “everything small is everywhere”, as traditionally asserted for these life forms [5]. This perception was primarily attributed to the small size of their propagules, whether asexual or sexual, which are easily dispersed over long distances by wind currents [6,7,8,9,10], and to their ability to tolerate extreme physiological stress [11]. However, recent advances in molecular techniques have unveiled that the occurrence of endemic species in this group is more prevalent than previously assumed [12,13,14,15,16,17,18]. These studies revealed that, despite the higher occurrence of widely distributed species in lichenized fungi compared to other groups such as animals or plants, lichens can have more constrained distribution areas than previously thought. Although the causes for this may be multifaceted and not thoroughly studied, these could include, in addition to limitations in dispersal, the ecological specialization of species concerning their habitat [19], recent speciation events [16], or a high degree of specialization concerning the photobiont [20]. Additionally, specific regions, such as certain island archipelagos (including Macaronesia), appear to exhibit higher rates of endemism compared to other areas [1].

Macaronesia constitutes a geographical region comprising five volcanic archipelagos situated in the Atlantic Ocean, namely the Azores, Madeira, Selvagens, Canary Islands, and Cape Verde, as well as a portion of West Africa referred to as the Macaronesian Enclave [21]. This region is distinguished by notably high levels of biodiversity and endemism across a diverse range of organisms [22] and is an integral component of one of the 36 World Biodiversity Hotspots, the Mediterranean Basin [23,24], in which the Canary Islands hold a significant position [25,26]. The lichen biota (including lichenicolous fungi) of the Canary Islands is rich, with approx. 1800 species listed for an area of just 7447 km^2^ [27]. However, the vast array of diverse ecosystems present on the islands coupled with the influence of anthropogenic pressure means that comprehensive understanding of their biota remains a distant prospect.

Human activities have introduced disruptions to ecosystems worldwide, and the Canary Islands are not an exception. After the Hispanic colonization in the 15th century, the natural landscape underwent increasing transformation due to the need for land for human settlement, agriculture, and livestock farming. This transformation escalated significantly with the tourist boom that emerged in the islands in the early to mid-20th century, continuing to the present day. This has turned the Canary Islands into one of the primary destinations for mass tourism in the European context, attracting around 16 million tourists annually [28]. As a result of this substantial human impact and pressure, there is an imperative and pressing need to study biodiversity, especially those organisms that have traditionally been neglected, like lichens. This is necessary and urgent to prevent their potential extinction without even being known. In this context, La Isleta, which is on the Island of Gran Canaria, serves as a clear example. It is a volcanic islet that has become a peninsula due to coastal sedimentation processes, allowing the creation of a sandy isthmus connecting it to the main island [29]. Following its isolation in the aftermath of the conquest of the original city of Las Palmas, and due to challenging environmental conditions (annual precipitation less than 160 mm, high insolation, volcanic terrain unsuitable for cultivation, etc.), its natural habitat remained well-preserved until the late 19th century. This changed with the construction of the Port of La Luz and Las Palmas (the largest and most significant in the Canary Islands), its subsequent expansions, and the associated urban growth. A substantial portion of La Isleta has remained untouched by this development as it has been a military base since the early 20th century, although it has been impacted by subsequent military maneuvers. Recently, the Gran Canaria Island Council (Cabildo de Gran Canaria) commissioned a study to highlight the natural values of this area, which has been declared a protected natural space in an effort to preserve these unique and increasingly scarce ecosystems in the island. Among the 80 lichen species found [30], there is a notable presence of specimens belonging to the genus *Xanthoparmelia* and assigned to the *Xanthoparmelia subramigera* group. However, the identity of the collected samples has not yet been confirmed, and we suspected that they could belong to a yet undescribed species.

*Xanthoparmelia* (Vainio) Hale is an ultradiverse lichen genus, currently comprising more than 800 recognized species, thereby ranking as the genus with the highest number of species among lichenized fungi [31,32,33]. The genus is widespread throughout the world but has two main centers of diversity: Australia and the Cape Region of South Africa [34]. The majority of species within this genus are typically found on siliceous rocks in arid, semi-arid, and Mediterranean climates, characterized by dry conditions and ample sunlight [35,36,37]. It displays a considerable morphological and chemical variation with more than 40 chemosyndromes represented [38]. These characters have traditionally been used to distinguish species [31,35,37,39]. Nonetheless, some studies have identified a correlation between morpho-chemical variation and the macro and micronutrient content of the thallus, hinting at the possibility that environmental factors could introduce complexity into the patterns of variability in *Xanthoparmelia* [40]. Consequently, it becomes imperative to incorporate molecular studies to contribute to a better understanding of the species concept within this genus and the relationships among these species. 

The present study aims to clarify the systematic position and biogeographic relationships of the *Xanthoparmelia* specimens found in La Isleta. The diagnostic morphological characters and the taxonomic status of this taxon will be evaluated using an integrative taxonomy approach, incorporating information from different sources: DNA sequences, morphological, chemical and macroclimatic data.

## 2. Materials and Methods

### 2.1. Morphology and Anatomy

The morphology of the lichen specimens was examined using a Leica ZOOM 2000, Buffalo, NY, USA). Sections for anatomical examination were made by hand and were studied under an Olympus CX33 (Shinjuku, Japan) equipped with a Motic camera Moticam S12 (Hong Kong, China). Color reactions (spot tests) were made using standard methods (Orange et al., 2001 [41]). Chemical constituents were identified by thin-layer chromatography (TLC) using standard methods and solvent systems A and C [41,42]. Specimens are deposited in the institutional Herbarium of the University of La Laguna (TFC Herbarium).

### 2.2. Molecular Analysis

#### 2.2.1. DNA Isolation, Amplification, and Sequencing

Small thallus fragments were removed with the help of forceps to avoid areas with obvious damage or presence of superficial epiphytes. Genomic DNA was purified from 10 mg samples with the E.Z.N.A DS Plant DNA Kit (Omega Bio-Tek, Atlanta, GA, USA) following the manufacturer’s protocol. Previously, samples were pulverized by vigorous shaking (5 m/s; 30 s) using 2 mL Lysing Matrix-A tubes and a FastPrep-24 System (M.P. Bio-medicals, Irvine, CA, USA). The three following genetic loci were PCR amplified: nuclear *ITS* rDNA using primer ITS1f [43] and ITS4 [44]; *nuLSU* rDNA using primers AL2R [45] and LR6 [46] and *mtSSU* using primers mrSSU1 [47] and mrSSU7 [48]. PCR reactions were carried out in a total volume of 20 μL containing 5 μL of diluted template DNA (0.4 ng/μL), 2 μL of each primer (2 μM), 2 μL of dNTPs (2 μM), 0.2 μL of Phire^®^ Hot Start II DNA Polymerase (ThermoFisher Scientific, Waltham, MA, USA), 4 μL of 5xPhire Reaction buffer (ThermoFisher Scientific, Waltham, MA, USA), and 4.8 μL of distilled water. The amplification programs were as follows: initial denaturation at 98 °C for 30 s; 30 amplification cycles with denaturation at 98 °C for 10 s, primer annealing during 10 s, and extension at 72 °C for 20 s; final extension at 72 °C for 30 s. Annealing temperatures were 56 °C, 61 °C, and 53 °C for *ITS* rDNA, *nuLSU,* and *mtSSU* loci, respectively. PCR products were run in 1.5% agarose/TBE gels stained with GelRed (Biotium, Fremont, CA, USA). Amplicons were enzymatically cleaned with ExoCleanUp (VWR) and sequenced by the Sanger method in an external service (Macrogen Inc., Seoul, Republic of Korea).

#### 2.2.2. Sequence Alignment and Phylogenetic Analysis

To explore the phylogenetic placement of the new species within the *Xanthoparmelia subramigera* group, we generated phylogenetic trees based on both Bayesian Inference (BI) and Maximum Likelihood (ML). In these trees, we included the sequences of the new species, sequences from the 18 specimens of the group available in GenBank [34,49], and sequences of *Xanthoparmelia hueana*, *X. patagonica,* and *X. subruginosa*, which were used as outgroups based on the results of [34]. GenBank accession numbers and specimen metadata are listed in Table 1. The sequences were aligned using MAFFT v7.450 [50], as implemented in Geneious Prime^®^ v2023.2. We set the following parameters: the FFT-NS-I ×1000 algorithm, a gap open penalty of 1.53, the 200 PAM/k = 2 scoring matrix, and an offset value of 0.123. We used MrBayes 3.2.7 [51,52] to calculate the Bayesian tree. The MrBayes analysis started with a random tree and two simultaneous, parallel four-chain runs were executed over 10^7^ generations, and they were sampled after every 1000th step. The first 20% of the data was removed as burn-in, and the 50% majority-rule consensus tree was calculated from the remaining trees. We used RAxML 8.2.12 [53] to calculate the Maximum likelihood tree and performed 1000 rapid bootstrap pseudoreplicates to evaluate nodal support. We used the GTRGAMMA substitution model. Nodes with posterior probabilities equal to or higher than 95% and with bootstrap values equal to or higher than 70% were considered to be significantly supported. Both programs were run in the CIPRES Science Gateway [54].

### 2.3. Dating Analysis

Speciation events in the *Xanthoparmelia subramigera* group were time-framed using secondary calibration. Specifically, we based the time-calibration on the time tree published by [34]. This tree was inferred from a 9-loci dataset comprising 124 species of the genus worldwide. We conducted the analysis in BEAST v.1.10.4 [55]. Different combinations of clock and tree models were explored, and, after Bayes Factors comparisons, we implemented a strict clock and the birth-death tree prior. We used the 10 Ma time estimation for the most recent common ancestor of the *Xanthoparmelia subramigera* group (8.55–11.52 Ma, 95% HPD) inferred by [34] to calibrate the tree. This calibration was implemented using a normal prior (mean = 10, stdev = 0.9). We used the substitution models inferred by the model selection tool in IQ-TREE 1.6.12 [56,57], which were TrNef+G for the *ITS*, K80+I for *nuLSU* and GTR+G+I for *mtSSU*. The analysis was run for 10^9^ generations, sampling every 10^5^. Chain convergence was checked using Tracer 1.7.1 [58], and the first half of the trees were discarded as burn-in. The remaining 5000 trees were passed to TreeAnnotator [59] to generate a maximum clade credibility tree.

### 2.4. Macroclimatic Niche Characterization

As an additional source of information, we sought to determine whether the new species experienced different macroclimatic conditions in Macaronesia than those of closely related taxa in mainland Africa and the type locality of *Xanthoparmelia subramigera*, which is the Rainbow Falls in Hawaii. For this, we downloaded georeferenced occurrence records of the *Xanthoparmelia subramigera* group from sub-Saharan Africa (https://doi.org/10.15468/dl.jufvd2, accessed on 31 December 2023) from GBIF and added an additional locality corresponding to the Rainbow Falls. Raw GBIF data can contain records that are imprecise, duplicated, or of a dubious nature [60]. A series of pre-processing steps were taken to detect and eliminate as many of these records as possible. More precisely: (1) only records associated with specimens preserved in herbaria were included in the analysis, (2) non-georeferenced records were dropped, (3) when multiple records of a given species had the exact coordinates, only one was maintained, and (4) records that appeared on bodies of water were dropped. Once these steps were taken, the remaining records were assigned to the biogeographical regions defined by [61] for sub-Saharan Africa. Also, the geographic coordinates of the records were used to extract macroclimatic data. Specifically, we obtained values for five macroclimatic variables available in WorldClim [62]: solar irradiance, precipitation, average temperature, wind speed, and water vapor pressure. These variables were used instead of the more common bioclim variables of WorldClim, as those are based only on rainfall and temperature and do not include information on other factors that can be important for lichens, such as irradiance or wind speed [63]. We downloaded all available layers of each variable (one for every month of the year) at a resolution of 2.5 arcminutes. Then, we merged them into yearly averages using the *calc* function of the *raster* R package 3.6.14 [64]. Variable values for each record were obtained using the function *extract* from the *raster*. Once extracted, we analyzed the macroclimatic data by principal component analysis (PCA). This was done using the *prcomp* function of the *stats* R package 4.2.2 [65].

## 3. Results

### 3.1. Sequence Alignment and Phylogenetic Analysis

In this study, we generated sequences for three different molecular markers for the new species. The DNA alignments had a length of 611 bp for the ITS, 915 bp for LSU, and 996 bp for mtSSU. The concatenated alignment had a length of 2522 bp, of which 114 were phylogenetically informative. The specimens of the new species formed a strongly supported clade in the phylogenetic tree (Figure 1). Interestingly, the new species appears inside a larger clade comprised exclusively by specimens collected in Kenya.

### 3.2. Dating Analysis

The chronogram inferred in the BEAST analysis placed the origin of the *Xanthoparmelia subramigera* group at 9.81 Ma (7.96–11.46 Ma HPD), in the Miocene (Figure 2). The new species diverged from its closely related Kenyan taxa at 3.52 Ma (2.2–4.87 Ma HPD), in the Pliocene.

### 3.3. Macroclimatic Niche Characterization

It is widely recognized that lichens have a strong association with microclimatic conditions. However, such data may not always be accessible, as in our case. Nevertheless, previous studies have reported that microclimatic conditions are partially correlated with macroclimate in most cases; see [66]. Therefore, approaches based on macroclimatic data can be informative.

The first two principal components of the PCA explained 75.52% of the variance (Figure 3). Samples from the Zambezian and Southern African regions overlapped greatly and appeared close to the two specimens from the Congolian region and the only specimen from the Somalian region (which includes Kenya). Our sampling locality in Macaronesia appeared between specimens belonging to the Southern African, Ethiopian, and Somalian regions. It appeared much closer to all African specimens, however, than to the type locality of *Xanthoparmelia subramigera* in Hawaii. *Xanthoparmelia ramosae* was segregated from the African records mostly by the second principal component. The variables with the highest contribution to this component were precipitation (52.47%) and temperature (37.48%). In comparison with the samples from the Congolian, Somalian, Zambezian, and Southern African regions, these specimens were collected in localities with less precipitation and higher temperatures.

### 3.4. Taxonomy

***Xanthoparmelia ramosae*** Pérez-Vargas & Blázquez *sp. nov.* (Figure 4).

Mycobank No: 851459

**Diagnosis**: Thallus adnate, upper surface yellow-green, emaculate, lower surface mid-brown, abundant isidia globose to cylindrical, simple to rarely branched and medulla containing usnic and fumarprotocetraric acids. Differs from the phenotypically similar *X*. *subramigera* because the former has much wider, contiguous, and imbricate lobes, white-maculate, generally larger and more loosely spreading thallus and molecular characters.

**Typus**: Spain, Gran Canaria Island, La Isleta, malpaíses centrales, UTM: 28°10′21″ N 15°25′48″ W, 20 m alt., NE exposure, on basaltic rocks in open *Euphorbia* shrubs, 18 September 2020 I. Pérez-Vargas (TFC Lich 17197 holotypus) (Figure 5).

**Etymology**: This new species is dedicated to the late Ana Ramos, a technician from the Cabildo de Gran Canaria. Without her perseverance and assistance, the development of this research project would not have been possible. We acknowledge her efforts in emphasizing the natural values of this protected area.

**Description:** Thallus saxicolous, foliose, adnate, up to 10 cm wide. Lobes are subirregular to sublinear, narrow and elongated, apically rotund, contiguous 1–2.5 mm wide but usually not more than 1 mm, and sparingly imbricate in the center of the thallus. Upper surface yellow-green, emaculate. Moderately dense to dense isidia in the center of the thallus, globose at first but then cylindrical, simple or rarely becoming coralloid, to 1 mm high, concolorous with the upper surface, and syncorticate. Soredia absent. Medulla white. Lower surface mid-brown not blackening. Rhizines sparse to medium, simple, concolorous with the lower surface, to 1 mm long. Apothecia and pycnidia not seen.

Photobiont chlorococcoid, *Trebouxia*-like. Mature vegetative cells were predominantly spherical in shape, although oval, pyriform, and kidney-shaped variations were also common, ranging in size from 12.2–13.7 to 14.5–15.2 µm in diameter. The cell wall was variable in thickness, ranging from 1 to 3 µm, and had flat local thickening and irregular secretory spaces, giving a distinctive appearance to both young and mature vegetative cells. Occasionally, a thickened cell wall in the form of a cap is formed on one side of the vegetative cells. 

**Chemistry**: Cortex K+ yellowish, C−, KC+ yellowish, Pd−, UV−; medulla K−, C−, KC+ pinkish, Pd+ rusty orange, UV−. Usnic and fumarprotocetraric acids detected by TLC.

**Habitat and distribution**: So far, the new taxon has only been found in the type locality in La Isleta, Gran Canaria. *Xanthoparmelia ramosae* occurs on basaltic rocks, in “tabaibal”, a typical xerophytic shrubs community of the lowlands of Canary Islands relicts of the ancient xeric Tertiary period Rand flora. Less than 15% of its potential distribution on the islands currently survives; this significant reduction is primarily due to urban expansion and the occupation of coastal land for tourism development [67]. The vascular vegetation is scarce, with some species of *Euphorbia*, such as *E. lamarckii* and *E*. *aphylla*, *Lycium intricatum*, and some introduced species of *Opuntia*. Although the climatic conditions are challenging (high insolation, very low precipitation, high wind intensity), lichen communities can develop in NE-oriented areas and can become significant. As accompanying species, it is worth mentioning that various species of *Ramalina* (especially from the *bourgaeana* group), *Seirophora scorigena* (Ach.) Frödén, as well as several crustose species like *Diploicia canescens* (Dicks.) A. Massal., *Pertusaria etayoi* Pérez-Vargas & C. Hdez.-Padr., or various species of *Lecidea*, grow on the rocks.

**Notes**: *Xanthoparmelia ramosae* clusters within the *X. subramigera* group, a mostly tropical, little-studied group of the genus. This group of species shares the presence of cylindrical isidia, a lower surface that ranges from pale to dark brown, and fumarprotocetraric acid as the main secondary metabolite [34,35,49]. The new species is morphologically characterized by an emaculate and isidiate upper surface with narrow and elongated lobes that have a brown lower surface, and chemically characterized by the presence of usnic and fumarprotocetraric acid in the medulla. Among *Xanthoparmelia* species, over 600 have a yellowish-green upper surface, usnic or isousnic acid in the upper cortex, and cell walls containing *Xanthoparmelia*-type lichenan [31,38,68]. More than 250 yellow-green species have been reported in Africa [69]. Yet, in the Macaronesian archipelagoes, these species are poorly represented, with just 12 recorded species [70,71,72,73]. Furthermore, isidiate *Xanthoparmelia* species are distributed in boreal, temperate, and tropical regions. Nevertheless, they commonly occur in semi-arid to arid regions worldwide, especially on siliceous rocks, such as granite and sandstone [74]. In Macaronesia, only six other isidiate species grow: the common and widespread *Xanthoparmelia conspersa* (Ehrh. ex Ach.) Hale, *X. tinctina* (Maheu & A. Gillet) Hale, and *X. verrucigera* (Nyl.) Hale, are all isidiate species, but they are morphologically different and readily distinguished by the black lower surface and the chemical components of the medulla (stictic or salazinic acids). *Xanthoparmelia plitii* (Gyeln.) Hale has subirregular lobes, cylindrical, simple or coralloid isidia, and a brown lower surface, but has more loosely adnate thalli, the isidia are smaller and darkening, and it contains stictic acid complex in the medulla [35]. The Canarian endemic *Xanthoparmelia perezdepazii* Pérez-Vargas & C. Hdez.-Padr. may resemble this species due to the presence of isidia, but the lobules are much more rotund, contiguous, and imbricate; it grows in the high mountain of Tenerife, above 2000 m, and it exhibits a very different chemical composition due to the presence of constipatic and protoconstipatic acid [72]. *Xanthoparmelia subramigera* (Gyelnik) Hale, originally found in the Hawaiian Islands, is an isidiate species with similar chemistry; it has much wider, contiguous, and imbricate lobes, white-maculate, and a generally larger and more loosely spreading thallus [35]. In addition to *Xanthoparmelia subramigera* (*s*. *str*.), which appears to have a wide distribution spanning from Mexico to Kenya (as previously noted by [49], *X. ramosae* could be confused with other species of the X. *subramigera* group outside Macaronesia. The genetically closest species seems to be *Xanthoparmelia phaeophana* (Stirton) Hale. This species is common in southern Africa and is further distinguished genetically, morphologically, and chemically by its uniformly white maculate surface, very wide lobes (up to 10 mm), a lack of isidia, and having as its main secondary metabolites fumarprotocetraric, succinprotocetraric physodalic, as well as protocetraric, virensic, and caperatic acids [35]. Additionally, there are several genetically closely related specimens whose identity has not been confirmed, all originating from East Africa, and they are likely to be lineages or species awaiting formal description; the two separate lineages are denoted as *Xanthoparmelia aff. subramigera* and one as *Xanthoparmelia aff. krogiae* by [34]. *Xanthoparmelia krogiae* Hale & Elix is an Eastern African endemic characterized by a thallus with broader lobes, very small cylindrical isidia, and most notably, a distinctive pale salmon-yellow-colored medulla; chemically, it possesses protocetraric acid, succinprotocetraric, and fumarprotocetraric [35], making it easily distinguishable from the new species. Other species within the genus share an identical or very similar chemical composition. These include *Xanthoparmelia hypomelaena* (Hale) Hale, *X. novomexicana* (Gyelnik) Hale, and *X. protomatrae* (Gyelnik) Hale. The first lacks isidia and has abundant substipitate apothecia with a black lower surface [35]. *Xanthoparmelia novomexicana* is restricted to the southern United States and Mexico, lacks isidia, and substipitate apothecia are common; it is primarily found in the southern United States and Mexico [35,49]. Conversely, *Xanthoparmelia protomatrae* is a fertile species widely distributed that displays a white maculate upper surface and lacks isidia [35], thus posing challenges in rapid differentiation from the new species.

Regarding the photobionts of the new species, observations were made on different thallus samples, and no discernible differences were detected between them. The photobionts tended to be located very close to the hyphae within the photobiont layer. The interactions between the photobiont cells and the fungal hyphae exhibited characteristics of the “simple” type described by [2], without invaginations or haustoria. No auxospores or sexual reproduction were observed. The taxonomic classification of the *Trebouxiophyceae* is mainly based on a phylogenetic approach, and its delimitation on the congruence of 18S rDNA sequence data [75,76]. In the way it is presently characterized, the class *Trebouxiophyceae* comprises a diverse range of unicellular, colonial, and multicellular algae, which are mainly found in freshwater and terrestrial ecosystems [77,78]. This class is well known for its particular relevance as the major eukaryotic partner in lichen symbiotic associations, and *Trebouxia* stands out as one of the frequently encountered genera of coccoid algae within lichen thalli [79]. Owing to their simple morphology and small size, our understanding of the diversity within this class remains somewhat limited. It is very likely that many new taxa will be discovered, especially among the lichen photobionts, particularly within the “*Chlorella*-like” group of lichenized algae [75,76,80]. The use of molecular and ultrastructural methodologies will be essential to carry out further research and to determine species or OTUs (operational taxonomic units) within this family. Since we have not conducted genetic analyses of the photobiont, we cannot definitively confirm that its genus is *Trebouxia*. Therefore, we prefer to maintain it at the family level (*Trebouxiophyceae*) until further studies are conducted.

## 4. Discussion

In this study we provide the first example of a Macaronesia-Eastern Africa disjunction in lichen-forming fungi. This pattern has been found in *Xanthoparmelia ramosae*, an endemic species from Gran Canaria that has been newly described on the basis of molecular, morphological, chemical and macroclimatic data, and closely related species from Kenya. 

It is indeed intriguing that this new species from the Canary Islands appears within a larger clade composed exclusively of specimens collected in Eastern Africa. Generally, biogeographic relationships have received more comprehensive attention and are better studied in other taxonomic groups, such as plants, as opposed to lichens. Numerous molecular phylogenetic studies of some Macaronesian plant groups have offered valuable insights into the relationships of the endemism of this region. While in most cases, the sister group to the Macaronesian clade is typically of North African or Western European origin [81,82], the colonization of Macaronesia has proven to be intricate. Because of this complexity of the colonization scenarios within Macaronesia, assigning these islands to a particular mainland source area is problematic [83]. There are examples of connections with the Americas, South Africa or, as in our case, with Eastern Africa [84,85,86,87,88,89]. The reasons explaining this disjunction between Macaronesia and East Africa are not entirely clear. In many cases, the explanation has been attributed to the Relict theory: species with a widespread distribution during the Tertiary that later experienced fragmentation and reduction of their ranges due to post-Tertiary climate change. This, in turn, led to the spread of a drier climate in Africa during the late Miocene (and the formation of the Sahara Desert), eliminating populations between these two regions i.e., [84,85]. This theory appears to be the most accurate for explaining the existence of the Rand-Flora and the disjunct distributions between Macaronesia and East Africa of numerous plant genera; the distribution patterns of these genera seem to have been formed through a combination of climate-driven extinction and vicariance within a previously extensive distribution see [90,91,92,93]. The alternative theory posits that these distribution patterns are ascribed to long-distance dispersal events see [94,95,96,97]. This process has the potential to be significantly more intricate, constituting a blend of both vicariance and long dispersal events [83]. 

While understanding and accepting that successful, structured long-distance dispersal may be facilitated by considering the factors outlined by [98] (i.e., the age of the islands, continued volcanic activity, winds and marine currents, or the presence of emerged marine banks in past times that could act as stepping stones), the dating of the group’s origin and its divergence in the Miocene, along with the emergence of the new species in the Pliocene, is consistent with the relict theory. Instances of both cases in lichens can be found in the literature. Long distance dispersal appears to be the most plausible explanation, especially in the case of lichens that disperse through spores [99,100,101,102,103,104]. This process has been demonstrated in some Macaronesian species [12,105]. In the case of species that reproduce through isidia (as is our case), these are responsible for more local vegetative dispersal [104,106,107], making long-distance dispersal events more challenging. Vicariance might play a significant role in other lichen groups, i.e., [108,109,110]. If vicariance were applicable, one would expect the species to be present in suitable habitats in North-Central Africa, representing refugial populations from a broader past distribution. However, there are several issues that complicate attempting to prove this hypothesis. On the one hand, the desertification of the Sahara has occurred over the past 6.000 years. Although there is evidence of the existence of desert areas in North Africa since the late Pleistocene, the climate change that occurred after the mid-Holocene humid period led to the formation of the Sahara, the largest desert on Earth [111,112]. *Xanthoparmelia ramosae* diverged from its congeners in the Pliocene, so this abrupt climate change could have wiped out all those potential populations. On the other hand, there is a significant gap in lichenological knowledge not only in North Africa but across the entire continent, making it extremely challenging to hypothesize in this regard. Therefore, despite the absence of information about these populations, this theory cannot be entirely ruled out.

## 5. Conclusions

In this study we described an endemic species from Gran Canaria, *Xanthoparmelia ramosae*, that, to the best of our knowledge, constitutes the first example of a Macaronesia-Eastern Africa disjunction in lichen-forming fungi. Our results indicate that this species diverged from its closely related African taxa in the Pliocene, which, together with the fact that it reproduces by isidia, points to the Relict theory as the likely mechanism behind the disjunction. The large gap in lichenological knowledge on the continent, however, suggests that this should be taken with much caution.

## Figures and Tables

**Figure 1 jof-10-00166-f001:**
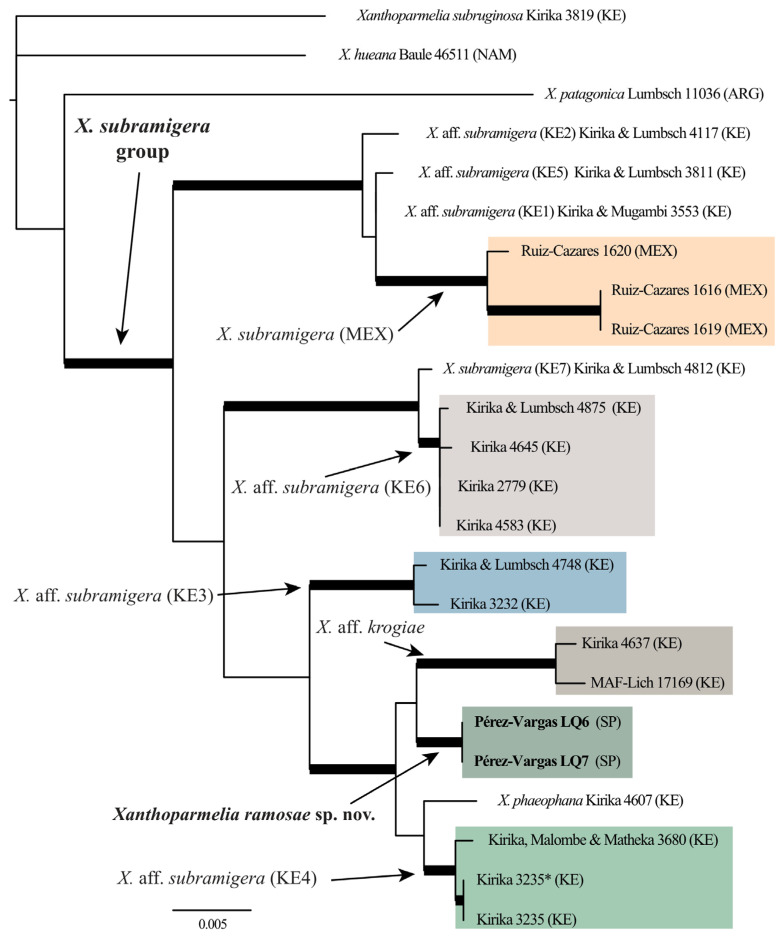
Phylogenetic tree of the *Xanthoparmelia subramigera* group inferred by Bayesian and Maximum likelihood analyses. Branches with posterior probability ≥95% and bootstrap values ≥70% are highlighted in bold. Species nomenclature follows [34].

**Figure 2 jof-10-00166-f002:**
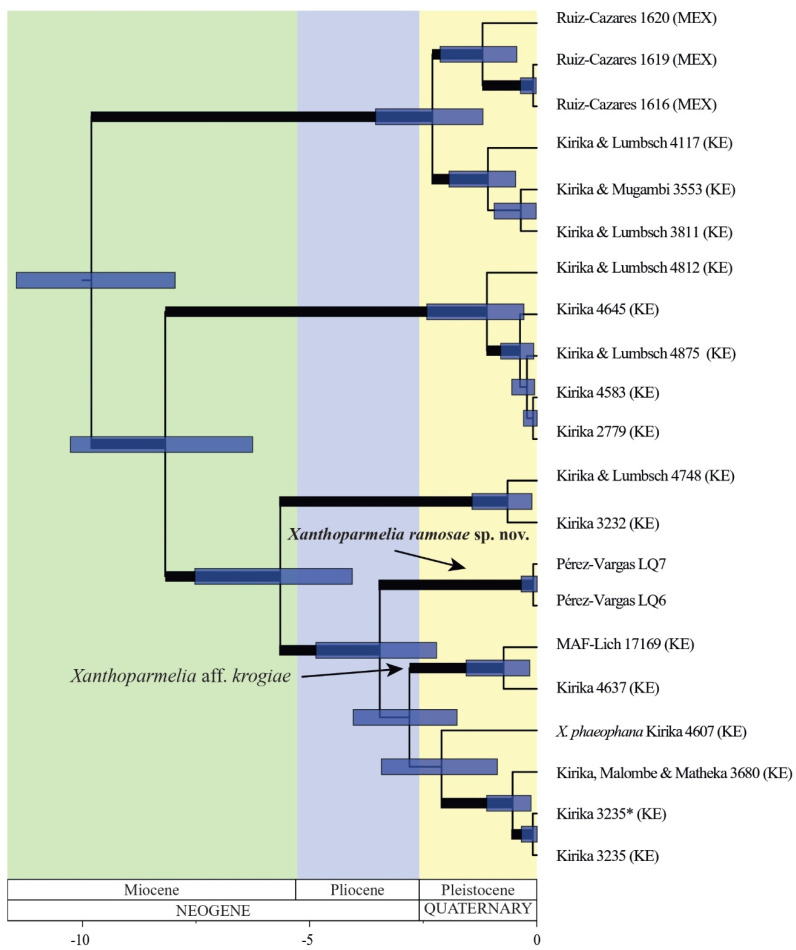
Chronogram obtained from the BEAST analysis depicting divergence times for the different taxa in the *Xanthoparmelia subramigera* group. Node bars show the 95% highest posterior density intervals (HPD). Branches with posterior probability ≥0.95 are highlighted in bold.

**Figure 3 jof-10-00166-f003:**
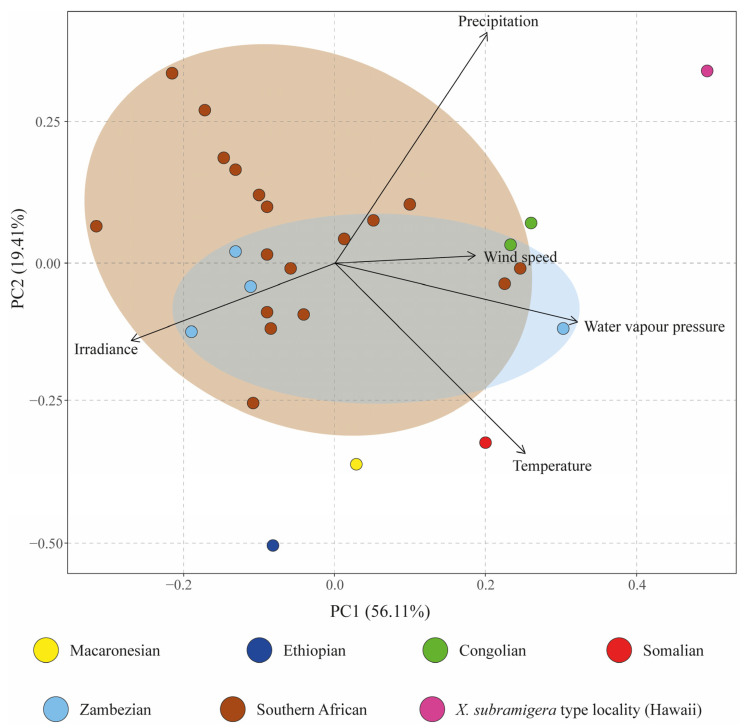
PCA plot computed from the macroclimatic variables extracted from the georeferenced GBIF occurrences. Sample color corresponds to the biogeographical regions defined by [61] for sub-Saharan Africa, the type locality of *Xanthoparmelia subramigera* in Hawaii and the Macaronesian region (type locality of *Xanthoparmelia ramosae*).

**Figure 4 jof-10-00166-f004:**
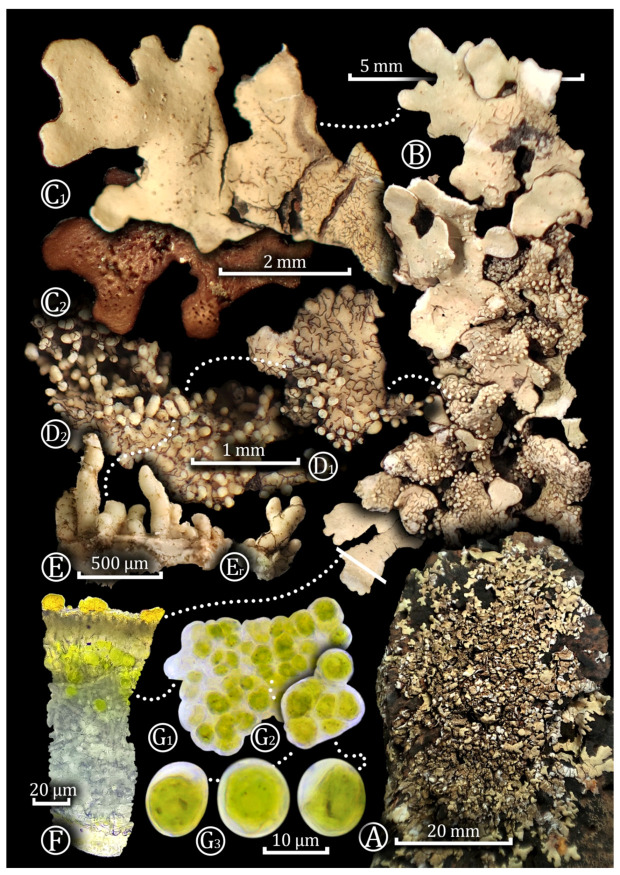
Morphology of *Xanthoparmelia ramosae*. A. Habit; B. Thallus lobes; C1–C2. Upper and lower surface; D1–D2, E–Er. Isidia; F. Transversal section through the thallus; G1, G2, G3. Photobiont cells.

**Figure 5 jof-10-00166-f005:**
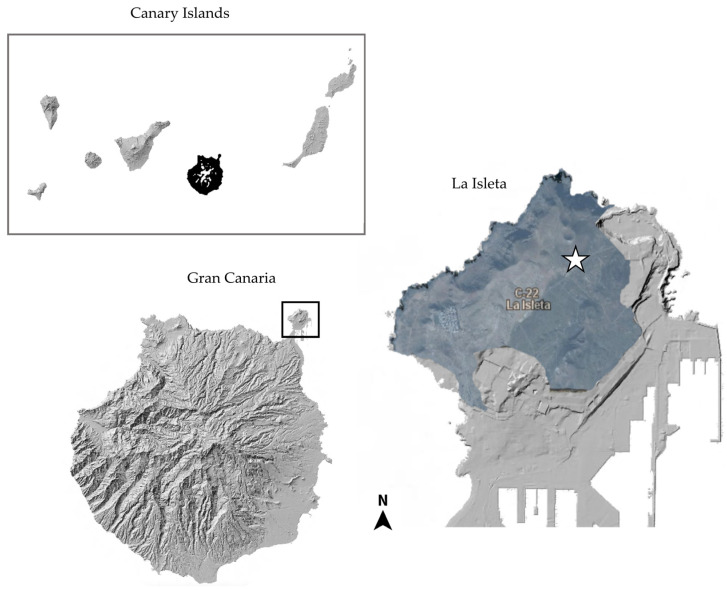
Star = Location of the *locus classicus* for the new species in La Isleta, Gran Canaria (Canary Islands, Spain).

**Table 1 jof-10-00166-t001:** GenBank accession numbers and associated mtadata of the specimens included in the phylogenetic analyses. Species nomenclature follows [34].

Species	Geographic Origin	Voucher	Source	*ITS*	*nuLSU*	*mtSSU*
*Xanthoparmelia hueana*	Namibia	Baule 46511	Leavitt et al. (2018) [34]	AY581090	AY578956	AY582326
*X*. aff. *krogiae*	Kenya	Kirika 4637	Leavitt et al. (2018) [34]	MG695502	-	MG695750
*X*. aff. *krogiae*	Kenya	MAF-Lich 17169	Leavitt et al. (2018) [34]	JQ912361	JQ912456	-
X. *patagonica*	Argentina	Lumbsch 11036	Leavitt et al. (2018) [34]	DQ980021	DQ923670	DQ923643
X. *phaeophana*	Kenya	Kirika 4607	Leavitt et al. (2018) [34]	-	MG695663	MG695817
*X*. aff. *subramigera* KE1	Kenya	Kirika & Mugambi 3553	Leavitt et al. (2018) [34]	MG695510	MG695610	MG695758
*X*. aff. *subramigera* KE2	Kenya	Kirika & Lumbsch 4117	Leavitt et al. (2018) [34]	MG695514	MG695615	MG695763
*X*. aff. *subramigera* KE3	Kenya	Kirika & Lumbsch 4748	Leavitt et al. (2018) [34]	MG695516	-	MG695765
*X*. aff. *subramigera* KE3	Kenya	Kirika 3232	Leavitt et al. (2018) [34]	MG695517	MG695617	MG695766
*X*. aff. *subramigera* KE4	Kenya	Kirika 3235	Leavitt et al. (2018) [34]	MG695520	MG695620	MG695769
*X*. aff. *subramigera* KE4	Kenya	Kirika 3235	Leavitt et al. (2018) [34]	MG695519	MG695619	MG695768
*X*. aff. *subramigera* KE4	Kenya	Kirika, Malombe & Matheka 3680	Leavitt et al. (2018) [34]	MG695518	MG695618	MG695767
*X*. aff. *subramigera* KE5	Kenya	Kirika & Lumbsch 3811	Leavitt et al. (2018) [34]	MG695521	MG695621	MG695770
*X*. aff. *subramigera* KE6	Kenya	Kirika & Lumbsch 4875	Leavitt et al. (2018) [34]	MG695523	MG695623	MG695771
*X*. aff. *subramigera* KE6	Kenya	Kirika 2779	Leavitt et al. (2018) [34]	MG695524	MG695625	MG695773
*X*. aff. *subramigera* KE6	Kenya	Kirika 4583	Leavitt et al. (2018) [34]	MG695525	MG695626	MG695774
*X*. aff. *subramigera* KE6	Kenya	Kirika 4645	Leavitt et al. (2018) [34]	MG695522	MG695622	-
*X*. aff. *subramigera* KE7	Kenya	Kirika & Lumbsch 4812	Leavitt et al. (2018) [34]	-	MG695628	MG695776
*X. ramosae* sp. nov.	Canary Islands	Pérez-Vargas LQ6	This study	OR957381	OR957383	OR957385
*X. ramosae* sp. nov.	Canary Islands	Pérez-Vargas LQ7	This study	OR957382	OR957384	OR957386
*X*. *subramigera* MX	Mexico	Ruiz-Cazares 1616	Barcenas-Peña et al. (2021) [49]	MW553779	-	-
*X*. *subramigera* MX	Mexico	Ruiz-Cazares 1619	Barcenas-Peña et al. (2021) [49]	MW553780	-	-
*X*. *subramigera* MX	Mexico	Ruiz-Cazares 1620	Barcenas-Peña et al. (2021) [49]	MW553781	-	-
*X*. *subruginosa*	Kenya	Kirika 3819	Leavitt et al. (2018) [34]	MG695533	MG695635	MG695783

## Data Availability

Data are contained within the article.

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
