# Peer review of "Patterns of Endemism in Lichens: Another Paradigm-Shifting Example in the Lichen Genus *Xanthoparmelia* from Macaronesia"

_jof, 2024, doi:10.3390/jof10030166_

Round 1
Reviewer 1 Report
Comments and Suggestions for Authors
The manuscript is very carefully prepared, in accordance with up-to date methodology, and well argumented.
Since the paradigm shift is a hypothesis with limited support, I would prefer a more informative title like “The new species X. ramosae indicates a new phytogeographical link, between the Canary Islands and East Africa”.
Introduction, line 71: Introduce here the subramigera-group and give references to the publication(s) where the group is defined (ref. 34 and 49? Note that Leavitt et al. calls the group Kenya 1.).
Line 123: Did you try a blast search on the Genbank website to gain information on the affinity of the new species? This will find the most similar sequences, and when the new species belongs to the subramigera group, the blast search will find suchsequences. A cladogram with a dozen or so most similar sequences from the blast search will show that the new species belongs in the subramigera group. That is more convincing then “tentatively”.
Comments on the Quality of English Language
Etymology: The text fragment “in memoriam” at the end of the paragraph is unusual. When you want to commemorate that Ana Ramosa died, add “the late” in front of her name on line 59. The whole paragraph needs some linguistic polishing.
Results, line 82: For a single plant use “shrub”, for the vegetation type “scrub” or “shrubs”.
Author Response
Thank you very much for taking the time to review this manuscript. Please find the detailed responses below and the corresponding revisions/corrections highlighted in the re-submitted files. With regard to the rest of the comments:
Since the paradigm shift is a hypothesis with limited support, I would prefer a more informative title like “The new species X. ramosae indicates a new phytogeographical link, between the Canary Islands and East Africa”.
In response to this argument, we would like to express our gratitude for the comment. However, we maintain our preference to retain the proposed title to emphasize not only the increasing frequency of describing new endemic species but also the fact that the revision of species previously thought to have extensive distributions is leading to the identification of lineages with restricted geographic distributions. By doing so, we aim to underscore the fact that the endemic element in lichens is far more prevalent than previously believed, and thus the paradigm of "everything small is everywhere" should be changed or at least reformed. We appreciate your consideration and valuable contributions.
Introduction, line 71: Introduce here the subramigera-group and give references to the publication(s) where the group is defined (ref. 34 and 49? Note that Leavitt et al. calls the group Kenya 1.).
The text has been changed accordingly to this comment (see "Notes")
Line 123: Did you try a blast search on the Genbank website to gain information on the affinity of the new species? This will find the most similar sequences, and when the new species belongs to the subramigera group, the blast search will find suchsequences. A cladogram with a dozen or so most similar sequences from the blast search will show that the new species belongs in the subramigera group. That is more convincing then “tentatively”.
Indeed, the first step before conducting the phylogeny was to perform a BLAST to identify the closest species, which in this case were Xanthoparmeia aff. krogiae (4630) and X. aff. subramigera (3235, 4645...); all these individuals are included in the analysis. In addition to this, in an exploratory manner, we generated a maximum likelihood tree of the entire genus (see attached at the end of the paper). This tree was based on the nine molecular markers used by Leavitt et al. (2018) and included a total of 192 specimens. We used this tree for two purposes. First, we made sure that the Xanthoparmelia subramigera group was recovered as monophyletic and that the new species appeared within this clade. Second, we selected three species that clearly appeared outside the group, Xanthoparmelia hueana, X. patagonica and X. subruginosa, as the outgroup for subsequent analyses. We confirmed that our new species falls within this group. Both the Blast and this prior analysis allowed us to determine the group to which the new species belonged, in addition to considering chemical and morphological characteristics. In the paper, we only display a portion of the tree to enhance data visualization.
Etymology: The text fragment “in memoriam” at the end of the paragraph is unusual. When you want to commemorate that Ana Ramosa died, add “the late” in front of her name on line 59. The whole paragraph needs some linguistic polishing.
We would like to thank the Referee for his/her comments. The text has been changed accordingly to all these comments.
Results, line 82: For a single plant use “shrub”, for the vegetation type “scrub” or “shrubs”.
We would like to thank the Referee for his/her comments. The text has been changed accordingly to all these comments.

Reviewer 2 Report
Comments and Suggestions for Authors
Interesting and well-performed study which discussed an uncommon type of endemism. The discussion looks substantiated and despite the true history of the taxon is disputable both the ways look reasonable. My concern is about the phylogenetic tree. The authors in fact represented only the Xanthoparmelia subramigera group with two other species used as an outgroup. A tree with a larger amount of species may change the view on the separation of the species. Using a software for delimitation of species (GMYC and bPTP) may substantiate the delimitation of taxa. Other notes please find in the text.

Author Response
Interesting and well-performed study which discussed an uncommon type of endemism. The discussion looks substantiated and despite the true history of the taxon is disputable both the ways look reasonable. My concern is about the phylogenetic tree. The authors in fact represented only the Xanthoparmelia subramigera group with two other species used as an outgroup. A tree with a larger amount of species may change the view on the separation of the species. Using a software for delimitation of species (GMYC and bPTP) may substantiate the delimitation of taxa. Other notes please find in the text.
We would like to express our gratitude to the referee for their comments, which undoubtedly have contributed to the improvement of the manuscript. About the concerns about the phylogenetic tree: We share the reviewer’s concerns about the effect that outgroup selection can have in tree topology. For this reason, as a prior step to the final phylogenetic analyses reported in the manuscript, we generated a maximum likelihood tree of the entire genus (see attached at the end of the paper). This tree was based on the nine molecular markers used by Leavitt et al. (2018) and included a total of 192 specimens. We used this tree for two purposes. First, we made sure that the Xanthoparmelia subramigera group was recovered as monophyletic and that the new species appeared within this clade. Second, we selected three species that clearly appeared outside the group, Xanthoparmelia hueana, X. patagonica and X. subruginosa, as the outgroup for subsequent analyses.Regarding the reviewer’s proposal about the use of automatic species discovery strategies, we agree that these methods can be of great help for taxonomists that seek to disentangle species boundaries in species complexes such as the Xanthoparmelia subramigera group. At this stage, however, we fear that the results of these algorithms would not be of much help, as a far more extensive sampling in continental Africa and elsewhere would be necessary for a proper species delimitation study in this group.
About other their specific comments:
Line 117. It has been an error; the McM7 marker has not been used, and this sentence has been removed from the text.
Line 136: What were the substitution models in these analyses. Were they the same as in the BEAST dating analysis? Please mention.
We used the GTRGAMMA substitution model. The text has been changed accordingly to this comment
Line 116. The text has been changed accordingly to this comment
Notes:
Line 116. The text has been changed accordingly to this comment
- Thank you very much for the comment, but we find it interesting to keep the paragraph as it provides information about the photobiont, and other referees have requested highlighting certain information from this paragraph
The scientific names that were not in italics have been italicized, and the minor typos pointed out by the referee have been corrected.

Reviewer 3 Report
Comments and Suggestions for Authors
I enjoyed reading the work, it is interesting to me. I found some formal mistakes like the lack of the abstract within the main text. Found my detailed comments in the attached document.

Author Response
We would like to express our gratitude to the referee for their comments, which undoubtedly have contributed to the improvement of the manuscript. With respect to the specific comments from the referee, we would like to make the following remarks:
I want to point few aspects that can be improved, as for example the quality of the images. Specially Image 1 can be improved to be more attractive (adjust the scaling, fonts etc.)
We have attempted to enhance the quality of Image 1 in accordance with the referee's comments.
L13-14 Abstract and Keywords missing.
When I received the invitation to review the article there was an abstract, however there is not in the main text. I will copy it here:
“It has long been assumed1 that lichen-forming fungi have very large distribution ranges and that endemic species are rare in this group of organisms2. This is likely a consequence of the “everything small is everywhere” paradigm that has been traditionally applied to cryptogams. However, the description of numerous endemic species in the last decades, many of them in oceanic islands, is challenging this view. In this study we provide another example, Xanthoparmelia ramosae, a species that is described here as new to science on the basis of molecular3, morphological, chemical and macroclimatic data. The new species is endemic to the island of Gran Canaria but clusters into a clade composed exclusively of specimens collected in Eastern Africa, a disjunction that is here reported for the first time in lichen-forming fungi. Through the use of dating analysis, we have found that Xanthoparmelia ramosae diverged from its closely related African taxa in the Pliocene. This result, together with the reproductive strategy of the species, points to the Relict theory as a likely mechanism behind the disjunction, although the large gap in lichenological knowledge in Africa makes this possibility hard to explore any further.”
Check the grammar, I am not sure if it is correct. Checked.
Instead of write molecular, you can specify in the abstract the number of loci analyzed, it can be very informative. The text has been changed accordingly to this comment.
We do not understand what may have happened with the abstract. This issue has occurred with some referees, while with others it has not, and the abstract also appears when we download the document from the figure's website. It must be a computer issue, and even though we are not responsible, we apologize for any inconvenience. We are including the abstract (and the rest of the paper, see attached) for your review
L71-72 Are you referring to the new species X. ramosae? In this case change to “suspected”. Since you already confirmed as a new species. The text has been changed accordingly to this comment.
L 89 I like the terminology integrative taxonomy to include morphology, chemistry, ecology, distribution, molecular characters and additional lines of evidence. The text has been changed accordingly to this comment.
L 92 Even if you explained properly in the introduction where you collected the samples, I consider interesting to add the sampling site in MM, maybe including a figure showing La Isleta within Gran Canaria Island, GPS coordinates, and whatever relative to the collection methodology.
The characteristics of the sampling locality, including the UTM coordinates, are briefly described in the section on the type locality. We have added a figure (Fig. 5) in accordance with the referee's comments
L 111 Check the space after reference. checked
L 170 “.. by Linder et al. [61] for sub-Saharan Africa..” write the name before the reference.
Although we consider the citation format you recommend to be more informative, in this instance, we are constrained by the journal's guidelines, which do not permit such an approach
The Line numbers are not continuous, it starts again from 1 in the results Section.
L 13 Species nomenclature follows [34].→Specify it in the text, “previous studies”, write the name of the author... L 43 the biogeographical regions defined by [61] for sub-Saharan Africa.
Similar to the previous case, although we deem the citation format you propose to be more informative, in this instance, we are bound by the guidelines of the journal, which preclude such an approach
L 53 italics. Checked
L 71 Are you sure that the photobiont is Trebouxia? Asterochloris and VUlcanochloris are pretty similar and without specific photobiont sequencing is hard to identify even at genus level. If there is any doubt, I suggest to modify it to Trebouxiophyceae.
Given that we have not conducted genetic analyses of the photobiont and, as the referee suggests, there are several genera of algae within this family that are very similar, we choose to classify the photobiont at the family level as the referee suggested.
L 154 Figure 4 Looks absolutely wonderful.
The discussion is brief and straightforward, I don ́t have comments trying to improve it. The conclusions are elegant and well supported.

Reviewer 4 Report
Comments and Suggestions for Authors
The paper is well written with a good English knowledge and very wide citation of references, furthermore the illustrations are of good quality.
Molecular genetic and statistical methods are applied during the research.
The authors were going to establish the relation of the investigated taxon from the Canary Islands to Xanthoparmelia subramigera. They found that it is related the closest to a part of the samples X. aff. subramigera in Kenya. However they do not treat this taxonomic-nomenclatural question fully.
An emendation of X. ramigera would be necessary for the remaining part of the species.
While they separate a part of X. subramigera specimens from the original species with a wider species concept, they do not handle the remaining part of the species (though mention supposed morphological differences) and neglect the possibility of their species actually being X. subramigera s.str. The original description is from coastal rocks of Hawaii Island (Rainbow Falls). No original material (e.g. fresh material from the type locality) was investigated in the current molecular genetic study, neither in previous cited sources.
Furthermore under Notes the species could be compared to other species with the same chemical content (e.g. X. protomatrae).
The macroclimatic analysis is valuable, however microclimatic conditions are more important for cryptogams. How the macro- and microclimate of type locality of basaltic rocks in open Euphorbia shrubs in Gran Canaria compares to those of rocks at Rainbow Falls? Is it possible to know anything about the conditions of habitats where the relevant Kenyan specimens were collected?
Other formal remarks:
Abstract is missing from the manuscript.
An addition of country abbreviations would be useful to add in Fig.1 and Fig. 2 legends.
Check some Latin names of taxa in the References and change to italics.
The manuscript is highly valuable, but additional work is necessary before its publication.
Author Response
The paper is well written with a good English knowledge and very wide citation of references, furthermore the illustrations are of good quality.
Molecular genetic and statistical methods are applied during the research.
The authors were going to establish the relation of the investigated taxon from the Canary Islands to Xanthoparmelia subramigera. They found that it is related the closest to a part of the samples X. aff. subramigera in Kenya. However they do not treat this taxonomic-nomenclatural question fully.
An emendation of X. ramigera would be necessary for the remaining part of the species.
While they separate a part of X. subramigera specimens from the original species with a wider species concept, they do not handle the remaining part of the species (though mention supposed morphological differences) and neglect the possibility of their species actually being X. subramigera s.str. The original description is from coastal rocks of Hawaii Island (Rainbow Falls). No original material (e.g. fresh material from the type locality) was investigated in the current molecular genetic study, neither in previous cited sources.
Furthermore under Notes the species could be compared to other species with the same chemical content (e.g. X. protomatrae).
The text has been modified in accordance with this comment.
The macroclimatic analysis is valuable, however microclimatic conditions are more important for cryptogams. How the macro- and microclimate of type locality of basaltic rocks in open Euphorbia shrubs in Gran Canaria compares to those of rocks at Rainbow Falls? Is it possible to know anything about the conditions of habitats where the relevant Kenyan specimens were collected?
We share the reviewer’s concerns about microclimatic conditions being more important for lichens than macroclimatic ones. This analysis was intended as an additional source of information to describe Xanthoparmelia ramosae as a new species under an integrative taxonomy approach, alongside molecular and phenotypic data. We agree that it would have been more appropriate to perform the analysis using microclimatic data but, unfortunately, this information is not available. We believe that the inclusion of the analysis using macroclimatic data is still useful, as previous studies have reported that microclimatic conditions partially correlated with macroclimate in most cases (see Gunderson et al., 2018).
Regarding the question about the Kenyan specimens, these were studied by Leavitt et al. (2018) and they did not provide geographical coordinates nor precise locality descriptions that could be used to georeference the specimens. For this reason, it was not possible to use the putative species proposed by these authors as study groups in the PCA. The suggestion about comparing the type locality of the new species with that of Xanthoparmelia subramigera in Hawaii, however, is both interesting and feasible. We have re-run the analysis including this locality and updated the appropriate sections under materials and methods and results to reflect these changes
Other formal remarks:
Abstract is missing from the manuscript.
We do not understand what may have happened with the abstract. This issue has occurred with some referees, while with others it has not, and the abstract also appears when we download the document from the figure's website. It must be a computer issue, and even though we are not responsible, we apologize for any inconvenience. We are including the abstract (and the resto of the paper) for your review
Check some Latin names of taxa in the References and change to italics.
The text has been modified in accordance with this comment.

Round 2
Reviewer 4 Report
Please refer to the paper Gunderson et al. 2018 in the text - you mentioned in your answer - and insert it into the reference list.
Answers are accepted.
Author Response
Thank you very much for your comment. The referenced paper (Gunderson et al., 2018) has been incorporated into both the text and the bibliography.
